# A Curve Maybe to Narrow: Description of an Anomalous Course of the Right Coronary Artery

**DOI:** 10.3390/diseases9030062

**Published:** 2021-09-17

**Authors:** Guido Pastorini, Elisa Bertone, Alberto Talenti, Mauro Feola

**Affiliations:** 1Cardiology Department, Regina Montis Regalis Hospital, ASL-CN1, 12084 Mondovì, Cuneo, Italy; guido.pastorini@aslcn1.it (G.P.); elisa.bertone@aslcn1.it (E.B.); 2Radiology Service, Regina Montis Regalis Hospital, ASL-CN1, 12084 Mondovì, Cuneo, Italy; alberto.talenti@aslcn1.it

**Keywords:** multimodality imaging, cardiac-gated imaging techniques, coronary vessel anomalies

## Abstract

Congenital coronary artery anomalies are rare but well-described causes of chest pain and, in some cases, link to sudden cardiac death. With the spread of advanced imaging techniques, the number of incidental findings is staggering, but little information has been given in order to rule out potential malignant cases in symptomatic adult patients. Here, we describe a case of an anomalous **course** of the coronary artery **with an acute (<45°) take-off angle, as well as an *inter-arterial* course between a dilated ascending aorta and a dilated pulmonary artery, and** how we could manage this patient in our clinical practice.

A 48-year-old woman with hypertension, type 2 diabetes mellitus, history of breast cancer treated with radiation and chemotherapy presented with atypical chest pain. Physical examination was within normal limits. Left ventricular ejection fraction (LVEF), **obtained with transthoracic echocardiogram, proved to be 60%**. Due to the patient’s clinical pre-test probability and the referral atypical chest pain, a **coronary computed tomography angiography** (CCTA) 128-MDCT (preceded by infusion of metoprolol for pre-medication) was then requested in order to exclude a coronary artery atherosclerosis disease. **The echocardiography evaluation of the first tract of coronary arteries did not have the sufficient quality required to gain a diagnostic opinion.**

The patient underwent an adaptive sequential CCTA, and curved multiplanar reconstructions, maximum intensity projections, and a volume rendering technique were used to assess the coronary arteries.

Surprisingly, the calcium (Agatston) score was 0 in total, and no plaques were seen on the coronary arteries. However, an aberrant right coronary artery origin was noted **coming** from the lateral surface of the right coronary sinus with an acute (<45°) take-off angle and **an** *inter-arterial* course between a dilated ascending aorta and a dilated pulmonary artery (Figure 1 and Figure 2). 

In order to assess potential ischemia induced by dynamic compression provided by this anomalous course, we performed a functional non-invasive imaging stress test using an inotropic and chronotropic agent. The patient underwent a dobutamine stress echocardiography (SE) with an incremental dose from 5 up to 40 micrograms/kg/min, assessing during each phase: wall motion, systolic and diastolic function, TAPSE, MAPSE, global longitudinal strain (GLS), left ventricular dyssynchrony index (PSD) and post-systolic index (PSI). We achieved a heart rate of 85% of the maximal predicted heart rate for the patient’s age at the peak of the SE: no symptoms, ST-T segment down sloping or abnormal ventricular wall motion emerged. **An improvement of TAPSE (22 mm > 31 mm), stroke volume (59 mL**
**→ 88 mL), average global longitudinal strain (GLS −19.4%, PSD 59 ms, GLS right ventricle −18%**
**→ GLS −21.8%, PSD 48 ms, PSI 0, GLS right ventricle −24%) emerged after dobutamine stimulation** (Figure 3).

The negative dobutamine stress echocardiography, along with the absence of intramural course, allocated the patient in a class of low probability of sudden cardiac death caused by inducible myocardial ischemia.

Anomalous origin of the coronary artery is a rare (but not exceptional) condition of congenital heart disease [1], affecting about 1% of the general population [2]; it may lead to myocardial ischemia and consecutive arrhythmia with a significant risk of sudden death. However, **the** number of **anomalous origins of coronary artery is going** to rise due to the widespread of cardiac CT or CMR to evaluate symptomatic/asymptomatic patients.

The anatomic high-risk features known encompass origin arising from the wrong sinus of Valsalva, slit-like ostium, acute take-off angle, high take-off, intramural course and inter-arterial course (in particular, if in the presence of dilated aorta and pulmonary artery or with cardiac hypertrophy) [3].

The Eckart‘s observational study [4], obtained from a population of 23 million young US Army recruits in military training camps subjected to strenuous physical exercise for a duration of 2 months, calculated the risk of sudden cardiac death (SCD) in this population to be around 0.56 per 100 thousand cases. The evidence from the autopsy [4] highlighted how an anomalous origin of the left common trunk from the opposite sinus of Valsalva was the most involved anomaly in the occurrence of sudden death.

The mechanism linked to SCD could depend on hypoplasia of the coronary culprit, a kinking of the proximal segment of the ectopic coronary artery, an onset with an acute angle, or a coronary course between the aorta and the pulmonary, which could determine a compressive effect. CCTA allows us to assess high-risk anatomic features and plays a pivotal role [5,6] since it provides **exhaustive** images (thanks also to 3D reconstructions) that are better quality than Cardiac MRI. In order to **identify a** high-risk condition (excluding a potential mechanical compression), functional **imaging** cardiac testing **is advised**. **The stressor should reproduce a** real-life condition with maximal heart rate (this **is why a dipyridamole imaging test has not been indicated)** [6,7].

Since we were looking for a more objective evaluation (unlike from a simple oculometric evaluation) that would detect minimal alteration also (i.e., reduction in longitudinal left ventricular contractile function) [7,8], we adopted the speckle tracking analysis.

With a negative dobutamine SE, the patient was conservatively managed. Otherwise, in the case of a positive (or uncertain) result, consistent with the literature [6], a coronary angiography with intravascular ultrasound and functional test (for instance ergonovine) should have been performed in order to evaluate the possible degree of coronary stenosis and risk of inducible myocardial ischemia even if no atherosclerotic coronary stenosis is demonstrated. Although the absence of inducible ischemia during dobutamine infusion might be sufficiently safe, the coronary spasm in the absence of collateral coronary flow with consequent collapse and/or ventricular fibrillation during extreme exertion should be considered [6].

## Figures and Tables

**Figure 1 diseases-09-00062-f001:**
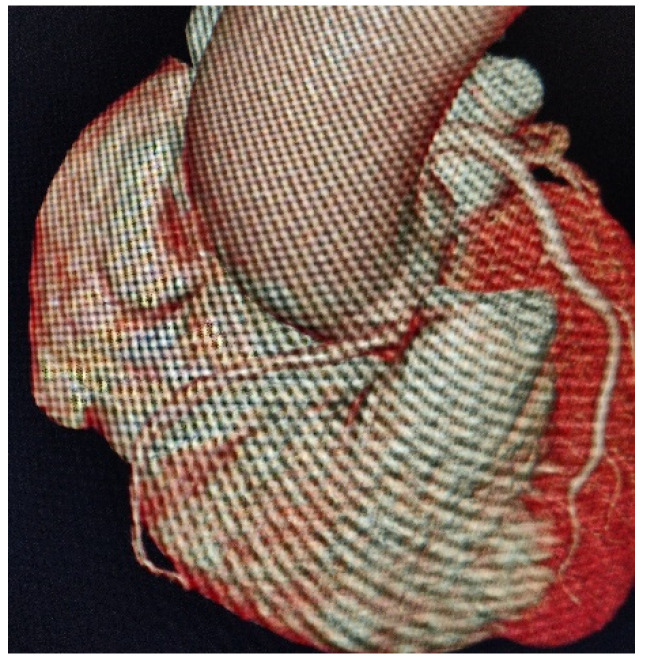
3D computed tomography (CT) volumetric visualization of inter-arterial course of right coronary artery.

**Figure 2 diseases-09-00062-f002:**
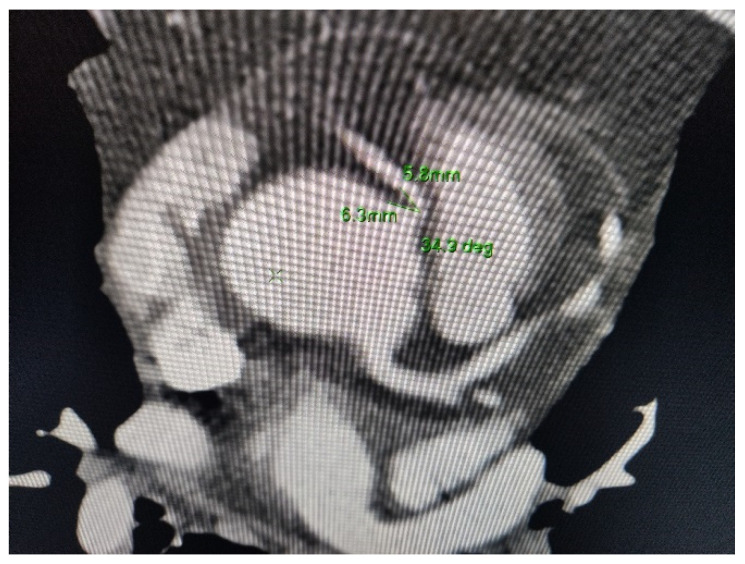
Maximum intensity projection visualization depicts the origin of a coronary artery with an acute angle (<50°), and then its passage between the aorta and the pulmonary artery, both dilated.

**Figure 3 diseases-09-00062-f003:**
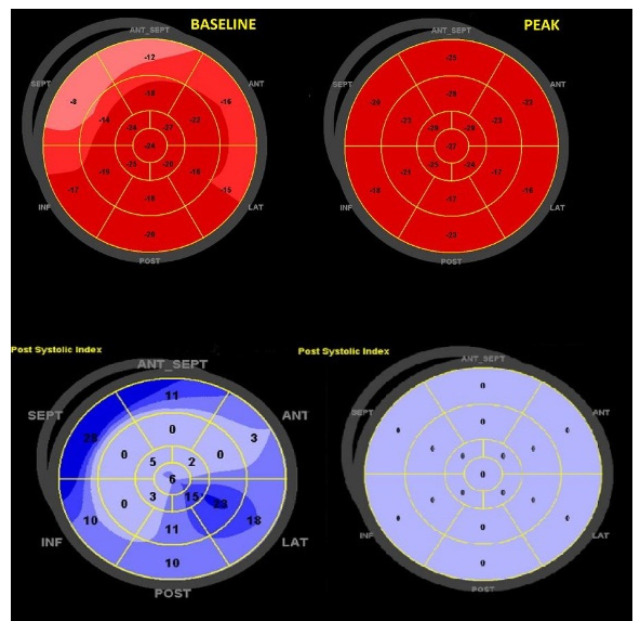
Comparison between baseline and peak GLS values. At baseline, there is a reduced longitudinal strain in the basal walls along with a moderate amount of post-systolic strain, typical of hypertension. At the peak, there is an increase in all segments (with now a PSI of 0), findings consistent with absence of ischemia and good response with stress. GLS = global longitudinal strain; PSI = post-systolic index.

## Data Availability

Not applicable.

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
