# Peer review of "A Curve Maybe to Narrow: Description of an Anomalous Course of the Right Coronary Artery"

_diseases, 2021, doi:10.3390/diseases9030062_

Round 1
Reviewer 1 Report
The authors reported the case of an anomalous origin of right coronary artery and their clinical practice on management. Please see my comments below;
*Please provide quantitative results instead of "normal". (such as LVEF?)
*Please correct all English grammar and typo errors.
*You should explain the abbreviation in the first place (such as CT).
*It seems very unfortunate that you couldn't see the great arteries anomalies by echocardiography, or did you? Please explain.
*I do not think that the finding anomalous coronary arteries incidentally is rare. Could you please more reference regarding this statement?
*Please focus on the anomalous of right coronary, not the other.
*Didn't you perform invasive coronary angiography? She had angina.
*Did you think that results of CCTA mislead you? Because curve area seems very small, and angle is too narrow.
Reviewer 2 Report
The authors reported an interesting case of an anomalous origin of the right coronary artery. I should not define it a proper "ACAOS" considering that the coronary artery origins from the right sinus. However it is interesting how the authors described the management of the patients. Few are the evidence reported in the literature. This report could improve it.
